# Molecular Tracing and Comparative Genomics Analysis of *Yersinia pestis* Strains Isolated from Wild Rodents in Yunnan Province in 2022

**DOI:** 10.3390/pathogens14121212

**Published:** 2025-11-28

**Authors:** Rong Yang, Fengyi Yang, Shanshan Dong, Haiyan Peng, Liyuan Shi, Peng Wang

**Affiliations:** 1School of Public Health, Kunming Medical University, Kunming 650106, China; running1678@163.com; 2Yunnan Key Laboratory for Zoonosis Control and Prevention, Yunnan Institute of Endemic Diseases Control and Prevention, Kunming 671000, China

**Keywords:** Yunnan Province, *Yersinia pestis*, plague, molecular tracing, comparative genomics

## Abstract

Three episodes of sylvatic plague occurred in Yunnan from April to November 2022, and nine *Yersinia pestis* strains were isolated. Whole-genome sequencing was performed on these isolates, and phylogenetic analysis based on SNP comparisons included 234 publicly available genomes from NCBI. All nine 2022 strains clustered within the 1.IN5 lineage, together with historical isolates from the Lijiang wild rodent plague focus. The Heqing strain HQ1 was most closely related to previous Heqing isolates, while seven Lijiang 2022 strains formed a monophyletic cluster with historical Lijiang strains; the remaining strain LJ4 was the closest relative to this cluster. Whole-genome comparison of HQ1 with historical Heqing strains revealed six SNPs and two indels. Of these, one nonsynonymous SNP and both indels—one being a deletion in the flagellin gene flgF—were located in coding regions. Comparison of the Lijiang strain 2022YL002 with historical local isolates identified ten SNPs and three indels. Five nonsynonymous SNPs were found in coding regions, including one at position 1566343 causing an amino acid change in the iron uptake regulator Fur, a virulence-associated mutation. All three indels were in coding regions. These findings confirm that the 2022 outbreaks originated from local plague reservoirs, while genetic differences indicate ongoing bacterial evolution. The results underscore the persistent activity of sylvatic plague in the Lijiang area and highlight the need for continued surveillance to prevent human spillover.

## 1. Introduction

Plague is a zoonotic disease caused by Yersinia pestis that originates from natural foci. It is characterized by acute onset, rapid progression, high mortality, and high transmissibility. The three major pandemics of plague worldwide have caused at least 160 million deaths [1], exerting tremendous impacts on the development and progress of human society. Yunnan has a long history of plague. According to literature records, the third global plague pandemic originated in Yunnan [2]. In 2005, an outbreak of human pneumonic plague occurred in Lijiang City, Yunnan Province, which attracted significant attention from relevant authorities and researchers. In the following year, the natural focus of plague in Lijiang was confirmed. Since then, the epizootic nature of plague among rodent populations in the Lijiang plague focus has remained persistently active [3]. Despite the absence of reported human plague in recent decades, *Yersinia pestis* continues to persist in wild rodents in regions such as Lijiang City and Heqing County. Its prolonged presence in animal reservoirs, coupled with antimicrobial resistance and multidrug resistance, underscores the need for ongoing surveillance due to the risk of sudden outbreaks [4,5].

Since the plague outbreak in 2005, relevant authorities have conducted multiple systematic investigations in the natural plague focus of Lijiang. The survey results indicate that the geographical range of the Lijiang plague focus has been progressively expanding. In 2017, two additional counties were identified as active epidemic areas—Heqing County and Jianchuan County of Dali Prefecture [6]. As of 2019, the natural plague focus in Lijiang spans mountainous areas at elevations ranging from 2300 to 3500 m above sea level, covering parts of Yulong County, Gucheng District, Heqing County, and Jianchuan County (only Jinhua Town). The affected area extends across 2 prefectures (cities), 4 counties (districts), 10 townships (towns), 31 village committees, and 71 villager groups, with a total area of approximately 1130 km^2^. The plague is spreading southward, with its epicenter moving southwest [7], posing greater challenges to surveillance and control efforts.

From April to November 2022, three episodes of sylvatic plague occurred sequentially in Yunnan Province (Figure 1A). One outbreak took place in Xiaomachang Village, Heqing County of Dali Prefecture, while the other two both occurred in Yulong County, Lijiang City, affecting areas of mountainous forests and cultivated lands in Jizi Village, Luzi Village, and Danhou Village (Figure 1B). These incidents were categorized as sylvatic plague [8]. During the epidemic response period, a total of 298 rodents were captured, including 274 live rodents and 24 dead rodents, with 346 flea specimens collected from these hosts. Nine strains of *Yersinia pestis* were isolated from the captured rodents and their associated fleas. Eight strains originated from Yulong County, Lijiang City, while the remaining one was isolated from Heqing County, Dali Prefecture. The isolates were designated as 2022YL002, 2022YL005, 2022YL006, LJ1, LJ2, LJ_zao, LJ3, LJ4, and HQ1, respectively. Notably, HQ1 was isolated from a live rodent, while all other strains were obtained from dead rodents (Table 1).

Molecular phylogenetic studies reveal that the most recent common ancestor of Yersinia pestis underwent two worldwide expansions within rodent and pika populations [1,9]. The first phylogenetic group 0.PE to emerge is represented by the Pestoides biotype, found in the Siberian jerboa, several vole species, and the Mongolian pika. Thousands of years after the formation of the 0.PE genovariants, the second wave of natural plague focus expansion emerged from the Tien Shan populations of the Altai marmot [10]. The ancestral form of this highly virulent genovariant was designated as 0.PE5. Based on its molecular characteristics, this variant is most closely related to the highly virulent “marmot-type” genovariants of the 0.ANT branch. The genovariants of the Mediaevalis biotype and the 2.MED group are not monophyletic; rather, they evolved in parallel from two distinct, marmot-derived ancestral genovariants: 2.ANT3 and 4.ANT1. During its westward expansion into the great gerbil populations of the Dzungaria region, the 4.ANT1 genovariant gave rise to 2.MED1, whereas its eastward dispersal into the Mongolian gerbil populations of the Khentii Mountains and Inner Mongolia, China, led to the evolution of the 2.MED3 genovariant [11]. The 1.ORI group emerged in the early 19th century, decades before the disease was first recorded as an epidemic in China, and was the key pathogen responsible for the global spread of all three waves of the third pandemic in 1894 [12].

Utilizing the genomic information of *Yersinia pestis* allows for tracing the origin of plague outbreaks. Through comparative genomic analysis, differences among various bacterial strains can be visually demonstrated [13,14,15]. This study performed whole-genome sequencing on nine *Yersinia pestis* strains isolated from three sylvatic plague outbreaks. Through molecular tracing methods and comparative genomic analysis, we aimed to elucidate the origin of the 2022 Yunnan sylvatic plague at the molecular level and to compare genomic differences between the newly isolated strains and previously identified plague isolates.

## 2. Materials and Methods

### 2.1. Experimental Strains

This study involved 9 Yersinia pestis strains isolated from the 2022 Yunnan sylvatic plague outbreak, which were provided by the Yunnan Institute of Endemic Diseases Control and Prevention. Detailed background information on strain isolation is presented in Table 1. Among the nine strains, five were isolated from Eothenomys miletus, one from Apodemus chevrieri, two from *Rattus tanezumi*, and the remaining one from Ctenophthalmus quadratus. (Notably, the primary hosts in the Lijiang sylvatic plague focus are Eothenomys miletus and Apodemus chevrieri, while both 2022YL002 and LJ3 were isolated from *Rattus tanezumi*.)

### 2.2. Pigment Adsorption Experiment

To assess the pigment adsorption capacity of the *Yersinia pestis* strain isolated from live rodents, the HQ1 strain was subjected to a pigment adsorption assay with the plague vaccine strain EV76 as a control. A bacterial suspension at 9 × 10^8^ CFU/mL was prepared in physiological saline using Falcon tubes, and serially diluted to concentrations of 10^7^, 10^6^, and 10^5^ CFU/mL. From each dilution, 100 μL aliquots were pipetted onto Congo red agar plates and uniformly spread with disposable “L”—shaped spreaders. Plates with well-dispersed and countable colonies were selected for further analysis. Following 3 days of incubation at 28 °C in a constant temperature incubator, colonies were examined for red pigmentation to determine their pigment adsorption capacity. Strains were classified as Pgm^+^ if ≥80% of colonies were red, Pgm^−^ if ≤ 20% were red, and Pgm^+/−^ for proportions between 20% and 80% [16]. This experiment was conducted in a Biosafety Level 3 (BSL-3) laboratory.

### 2.3. DNA Extraction

The 9 Yersinia pestis strains were individually inoculated into LB medium supplemented with rabbit hemolysate and cultured at 28 °C for 24 h. For Next-Generation Sequencing, DNA extraction was performed using the Qiagen DNeasy Blood & Tissue Kit (Qiagen, Hilden, Germany) according to the manufacturer’s protocols. For Third-Generation Sequencing, the Wizard Genomic DNA Purification Kit was employed following the manufacturer’s instructions. All DNA extractions were performed in a BSL-3 laboratory.

### 2.4. Sequencing

#### 2.4.1. Next-Generation Sequencing

The nine newly isolated Yersinia pestis strains were sequenced on an Illumina MiSeq 2500 platform. Paired-end sequencing libraries with an average insert size of 500 bp were constructed [17]. The raw sequencing data underwent quality control and filtering using FastQC (v 0.11.9) software. The filtered data were then assembled. The sequencing service was performed by Tsingke Biotechnology Co., Ltd. (Beijing, China) After obtaining clean reads, we assembled the filtered reads with the ABySS (v2.2.0), producing a FASTA file for downstream analysis.

#### 2.4.2. Third-Generation Sequencing

Following sample quality control, large-fragment DNA was enriched and purified using magnetic beads. The procedure included repairing damaged DNA fragments, performing end-repair and A-tailing of the DNA fragments, and conducting ligation reactions using adapters from the LSK109 ligation kit. The constructed DNA library was then quantified using Qubit, and sequencing was performed on the Nanopore platform. All aforementioned steps were conducted by Guangdong Meige Gene Biotechnology Co., Ltd, Guangzhou, China. After sequencing, clean reads were obtained. For pure third-generation sequencing data, assembly was performed using SMRT Link v5.1.0 software, whereas for hybrid assemblies requiring both data types, the Unicycler (v 0.4.8) software was used.

### 2.5. SNP Identification

The newly identified 9 Yersinia pestis strains, along with 234 publicly available genomic sequences of Yersinia pestis from the NCBI database were aligned to the reference strain CO92 (NC_003143.1) using BWA (v 0.7.17) for the detection of genome-wide single nucleotide polymorphisms (SNPs). The identified SNPs were combined, and those with coverage depth < 5× or located low-quality SNPs within 10 bp flanking regions were filtered out to minimize the impact of recombination events [18].

### 2.6. Statistical Analysis

A database was established using Excel 2003, and statistical analysis of the number of the identified SNP was performed with SPSS 26.0. After normality testing of the data, a *t*-test was applied to compare the number of SNPs between the newly isolated strains in 2022 and previously isolated strains, with a significance level set at α = 0.05.

### 2.7. Construct Phylogenetic Tree Based on Whole Genome SNP

A maximum-likelihood phylogenetic tree was constructed based on SNP sites identified in genomic sequence of 9 newly isolated Yersinia pestis strains from Yunnan in 2022 and 234 publicly available sequences (downloaded from https://www.ncbi.nlm.nih.gov/genome accessed on 9 March 2023). The CO92 strain was used as the reference for alignment, and representatives of the 0.PE7 lineage were included as the outgroup. The phylogenetic tree was visualized using FigTree v1.4.4 [19]. To achieve higher resolution, a minimum spanning tree (MST) was constructed using BioNumerics software (v 6.6) by performing “Advanced cluster analysis” for categorical data under the “Clustering” module [20]. The analysis included 9 Yersinia pestis strains newly isolated in 2022 along with 20 publicly available sequences previously obtained from Yunnan (see Table 2). A total of 20 publicly available sequences were isolated in 2006, 2014, 2017, and 2018 from locations including Yulong County and Gucheng District of Lijiang City, and Heqing County of Dali Prefecture. The sources of isolation were three rodent species (*A. chevrieri*, *Eothenomys* spp., *Rattus nitidus*) and two flea species (*N. specialis specialis*, *Ctenophthalmus quadratus*). The strain Yersinia pestis Z176003 (lineage 1.IN2; GenBank accession no.: NC_014029.1) was designated as the outgroup.

### 2.8. Comparative Genomics Analysis

VarScan (v 2.4.4) [21] software was employed to detect SNPs and Indels in sequences of *Yersinia pestis* strains. The software performs variant calling using BAM files generated from sequence alignment data. For each variant, parameters including read depth, number of supporting reads, mean base quality, and strand bias for each allele are reported. All parameter thresholds can be customized via option settings. During its execution, VarScan first processes the raw sequencing data by performing quality control to remove low-quality bases and sequencing errors, thereby improving overall data quality. Subsequently, the processed reads are aligned to the reference genome to obtain sequencing depth and base information for each genomic position. Finally, SNPs and insertions/deletions (indels) are detected based on the sequencing depth and base information.

## 3. Results

### 3.1. Results of the Pigment Adsorption Experiment

After 3 days of incubation, all colonies of HQ1 turned red on Congo red agar (Figure 2), indicating pigment adsorption capacity and thus classified as *Pgm^+^*. In contrast, the control strain EV76 showed virtually no color change, demonstrating absence of pigment adsorption ability and was designated as *Pgm^−^*.

### 3.2. SNP Identification of Yersinia pestis

The genomic sequences of the nine Yersinia pestis strains were assembled, yielding 173 to 189 discontinuous contigs with an average genome assembly size of 4.65 Mb and GC content ranging from 47.5% to 47.6%. Genome annotation for the sequences of the 9 strains isolated in 2022, along with 20 previously isolated strains from the sylvatic plague focus in Lijiang, Yunnan, was performed using the RAST server (https://rast.nmpdr.org/rast.cgi accessed on 10 September 2023). The gene count for these strains ranged from 4504 to 4653. Using the CO92 genome sequence as the reference strain, a total of 638 SNPs were detected among the 9 newly isolated Yersinia pestis strains in 2022, with the number of SNPs per strain ranging from 69 to 74 (mean: 70.89). Of these 638 SNP loci, 512 were located in coding regions (CDS), while 126 were in non-coding regions. Among the coding SNPs, 148 were synonymous and 364 were non-synonymous, resulting in a non-synonymous/synonymous SNPs ratio of 2.46 (Table 2 and Table 3). For the 20 previously isolated strains from the sylvatic plague focus in Lijiang, Yunnan, the mean number of SNPs per strain was 67.6, with a non-synonymous/synonymous SNPs ratio of 2.62 (Table 2 and Table 3).

### 3.3. Comparison of SNP Number Differences

Statistical analysis revealed significant differences (*p* < 0.05) in the total number of SNPs, synonymous SNPs, and non-synonymous SNPs between the newly isolated Yersinia pestis strains in 2022 and historically isolated strains in Lijiang and Heqing outbreaks. The newly isolated strains exhibited higher counts of total SNPs, synonymous SNPs, and non-synonymous SNPs compared to the historical strains. However, the ratio of non-synonymous to synonymous SNPs (nsSNP/sSNP) showed no statistically significant difference between the two groups (*p* > 0.05) (Table 3).

### 3.4. Yersinia Pestis Phylogenetic Tree

To clarify the phylogenetic position of the Yersinia pestis strains isolated from the 2022 Yunnan sylvatic plague outbreak within the global context, we constructed a phylogenetic tree based on 4665 SNPs, incorporating these nine newly isolated Yunnan strains along with 234 publicly available sequences. The latter included 20 strains previously isolated from the sylvatic plague focus in Lijiang, Yunnan. The nine Yersinia pestis strains isolated in 2022, along with previously isolated strains clustered within the 1.IN5 lineage (Figure 3). To more clearly illustrate the phylogenetic relationships among these strains, this study re-identified 67 SNPs shared by the 29 Yunnan sylvatic plague strains located on the 1.IN5 lineage branch. Using Z176003 (lineage 1.IN2, accession number: NC_014029.1) as the outgroup, a minimum spanning tree was constructed based on these SNPs (Figure 4).

The results revealed that seven Yersinia pestis strains isolated from the 2022 Lijiang sylvatic plague (2022YL002, 2022YL003, 2022YL005, LJ1, LJ2, LJ3, and LJ_zao) clustered with historically isolated Lijiang sylvatic plague strains, forming a distinct cluster named as Lijiang Cluster. Strain LJ4 showed the closest phylogenetic relationship to Lijiang Cluster, with 4 SNP differences between them. Meanwhile, the HQ1 strain isolated from Heqing County in 2022 demonstrated the closest phylogenetic relationship to Heqing Cluster with 4 SNP differences between them. Heqing Cluster comprised historically isolated strains from Heqing sylvatic plague in 2017.

### 3.5. SNP and Indel Variations

Comparison between the *Yersinia pestis* strains isolated from Heqing in 2022 and the strains isolated from Heqing previously revealed 6 SNPs and 2 Indels (Figure 5A). Among the 6 SNPs, 2 were located in intergenic regions and 4 were in coding regions. Among the 4 coding region SNPs, a non-synonymous mutation at position 597030 was identified within a gene encoding a component of a hydroxylated aromatic compound efflux system. The other three were synonymous SNPs located at positions 62127, 1565303, and 3828796, which are involved in the encoding of a hypothetical protein, β-N-acetylglucosaminidase, and flagellar P-ring protein (FlgI), respectively. The two Indels consisted of an insertion at position 1,875,996 and a deletion at position 3,831,996, which are involved in encoding λ-type phage Rac integrase and flagellar protein FlgF, respectively (Table 4). Comparison between the *Yersinia pestis* strains isolated from Lijiang in 2022 and the strains isolated from Lijiang previously revealed 10 SNPs and 3 Indels (Figure 5B). Among these 10 SNPs, 3 were located in intergenic regions and 7 in coding regions. Of the 7 coding region SNPs, five were non-synonymous mutations at genomic positions 1566343; 1856166; 3567279; 3931963 and 4069579, encoding iron uptake regulator Fur (associated with virulence genes), LysR-type transcriptional regulator, hypothetical protein, aspartate aminotransferase, and pentatricopeptide repeat protein family, respectively. The other two were synonymous mutations at genomic positions 36885 and 1439538, encoding methyl-accepting chemotaxis signal transduction protein and 2-succinyl-5-enolpyruvyl-6-hydroxy-3-cyclohexene, respectively. All three Indels were located in coding regions: two insertions at positions 740063 and 815034, encoding hypothetical protein and ABC transporter substrate-binding protein, respectively, and a deletion at position 3276758 with no annotated functional association (Table 5).

## 4. Discussion

### 4.1. Source of Sylvatic Plagues in Yunnan in 2022

We conducted phylogenetic analysis of Yersinia pestis strains isolated from the 2022 Yunnan sylvatic plagues, along with historically isolated strains from the Lijiang sylvatic plague focus and publicly available sequences from the NCBI database. The results indicated that all nine strains isolated from the three sylvatic plague outbreaks in Lijiang in 2022 clustered together with 20 historically isolated strains from the same sylvatic plague focus in Lijiang within the 1.IN5 lineage (Figure 3). This lineage, identified as a novel phylogenetic branch by Shi et al. research group [22] in 2021, comprises strains exclusively isolated from the Lijiang sylvatic plague focus in Yunnan. These findings indicate that all three sylvatic plague outbreaks in Yunnan in 2022 originated from local reservoirs rather than external introductions.

To further elucidate the phylogenetic relationships among the 29 strains within the 1.IN5 lineage, we re-identified SNPs between strains and constructed a minimum spanning tree. The results revealed that among the eight strains isolated from the two sylvatic plague outbreaks in Lijiang in 2022, seven clustered with strains historically isolated in Lijiang, forming a distinct Lijiang Cluster. The remaining strain (LJ4) was identified as the closest relative to the Lijiang Cluster. Meanwhile, the strain isolated from the 2022 Heqing sylvatic plague (HQ1) demonstrated the closest phylogenetic proximity to the Heqing Cluster formed by previously isolated Heqing plague strains.

Since 2006, the Lijiang plague focus has experienced plague outbreaks in eight different years up to 2022, involving a total of 27 distinct epidemic sites. Notably, in 2017, the scope of the plague focus expanded beyond Lijiang City (Yulong County and Gucheng District) to include Heqing County and Jianchuan County of Dali Prefecture [23,24]. Although the affected areas fall under two separate prefecture-level administrative regions—Lijiang City and Dali Bai Autonomous Prefecture—their actual geographical distribution spans only about 1000 square kilometers. Analysis of the population genetic structure based on a minimum spanning tree revealed that the *Yersinia pestis* population in this region exhibited a high degree of genetic stability, with well-conserved core genomes. Only a few strains exhibited relatively significant single-nucleotide polymorphism (SNP) variations, with the overall population forming a stable clonal complex. These genetic characteristics indicate that this area constitutes an independent natural plague focus, and the evolutionary relationships within the local *Yersinia pestis* population are consistent with a clonal reproduction model.

Previous investigations have identified Apodemus chevrieri and Eothenomys miletus as the primary reservoir hosts for Yunnan sylvatic plague [25]. However, during both plague outbreaks in April and November 2022, two Yersinia pestis strains were isolated from *Rattus tanezumi* (2022YL002 and LJ3). The minimum spanning tree analysis indicated that 2022YL002 and LJ3 shared an identical genotypes with strains isolated from wild rodents. This suggests that the infection of *Rattus tanezumi* likely resulted from cross-species transmission from wild reservoirs. The sylvatic plague focus in Lijiang is characterized by a mosaic landscape where residential areas, farmland, woodlands, and shrublands interlace closely. In this environment, the activity ranges of wild rodents and commensal rats exhibit significant overlap, creating opportunities for the transmission of plague from wild rodents to commensal species. *Rattus tanezumi* is a common species in the genus Rattus that frequently associates with human habitats. Theoretically, this proximity facilitate a transmission pathway from wild rodents to commensal rats and then to humans primarily via flea bites.

### 4.2. Comparative Genomic Analysis

In genomic evolution, genetic variation serves as the source of biological diversity, and natural selection acts upon this diversity [26]. The genetic diversity of the Yersinia pestis genome is primarily manifested through variations in SNPs, Indels, genomic rearrangements, and copy number variations. Compared to SNPs, Indel variations are more prone to cause frameshift mutations and alterations in variable number tandem repeats (VNTRs), thereby exerting more substantial impacts on gene function and corresponding phenotypic outcomes. We primarily compared SNP and Indel variations between the nine *Yersinia pestis* strains isolated from the 2022 Yunnan sylvatic plague outbreaks and previously isolated strains.

The newly isolated strains from 2022 exhibited higher total SNP counts, as well as increased numbers of synonymous SNPs (sSNPs) and non-synonymous SNPs (nsSNPs), compared to previously isolated strains, indicating a greater degree of genomic variation. However, the ratio of non-synonymous to synonymous SNPs (nsSNP/sSNP) showed no statistically significant difference between the two groups, with both ratios exceeding 1. This suggests that although the 2022 strains have accumulated more mutations, the relative proportion of non-synonymous to synonymous mutations remains stable. This pattern implies that positive selection pressure (e.g., adaptive evolution) may have maintained relatively constant levels during the long-term evolution of Yersinia pestis.

The strain HQ1, isolated from a live rodent in Heqing in 2022, exhibited 6 SNP and 2 Indel differences compared to previously isolated Heqing strains. Among these, a variant at site 3831996 is a deletion (loss of a single thymine base, Table 4) located within the gene region encoding the flagellar protein FlgF. This deletion induces a frameshift mutation that is likely to inactivate the gene. Although gene inactivation represents genomic decay, it can provide adaptive advantages at the species level by facilitating host adaptation and modulating virulence [27]. *Yersinia pestis* has already lost motility and does not produce flagellin. Most of its flagella-related genes have degenerated into pseudogenes due to the accumulation of inactivating mutations. The frameshift mutation in the *flgF* gene of strain HQ1 can be regarded as a recurrent inactivation event during evolution, further demonstrating the functional degeneration of the flagellar system in *Yersinia pestis*.

The Yersinia pestis strain 2022YL002, isolated from Lijiang in 2022, exhibited 10 SNP and 3 Indel variations compared to previously isolated Lijiang strains. Among these, the variation at position 1566343 was identified as a non-synonymous SNP, involving a base substitution from C to G. This mutation resulted in an amino acid change from histidine to aspartic acid in the encoded ferric uptake regulator Fur (Table 5), which is associated with iron acquisition regulation and contributes to the virulence of *Yersinia pestis*. A non-synonymous SNP refers to a single nucleotide mutation within the protein-coding region of a gene that results in an alteration of the amino acid sequence. This type of mutation may lead to functional changes in the corresponding protein [28], thereby potentially affecting gene function. Fur, as a global regulatory factor in bacteria, not only modulates iron uptake, storage, and metabolism but also confers stress resistance and pathogenic capabilities, enabling stable growth and reproduction in external environments [29]. In the Yersinia pestis genome, the iron uptake regulatory system is governed by the Fur protein. Under iron-replete conditions, Fur functions as a repressor that downregulates iron acquisition genes—including the yersiniabactin-encoding ybtT, catecholamine siderophore-related ybtE, and irp genes—thereby suppressing their transcription and facilitating preferential expression of virulence-associated genes. Conversely, during iron deficiency or limitation, iron uptake-related genes are preferentially expressed. This context-dependent gene prioritization mechanism enables Yersinia pestis to adaptively optimize its growth and reproduction within the host environment while concurrently enhancing its pathogenicity [30,31]. Consequently, the non-synonymous SNP at position 1,566,343 in the 2022YL002 genome resulted in an amino acid alteration in the virulence-associated Fur protein, thereby potentially affecting its function. However, further investigation is needed to elucidate the effect of this mutation on the virulence of *Yersinia pestis*.

## 5. Conclusions

In this study, we confirmed that two epizootic plague outbreaks in Yulong County, Lijiang City, and one outbreak in Heqing County, Dali Prefecture—all occurring in Yunnan Province in 2022—originated from the Lijiang sylvatic plague focus. The isolated strains were phylogenetically classified within the 1.IN5 lineage and constituted a genetically stable clonal cluster. A further comparison between the 2022 epidemic strains and previous strains revealed no variations significantly associated with virulence. It is noteworthy that the HQ1 strain, isolated during the 2022 Heqing outbreak, was again obtained from a live rodent host. Its epidemiological characteristics resemble those reported during the 2017 epizootic in the Heqing region. Such an epidemic pattern primarily involving live host animals exhibits significant stealth characteristics due to the absence of noticeable animal mortality, thereby substantially increasing the potential transmission risk of human plague. Moreover, the observed pathogen transmission from wild rodents to commensal rats in the Lijiang plague focus during 2022 further indicates an escalating risk of Yersinia pestis spillover into human populations in this region. Therefore, sustained enhancement of active surveillance and epidemiological early warning systems in the Lijiang and surrounding plague natural foci carries significant public health importance. However, as this study is based solely on bioinformatic analysis and lacks experimental validation, it has inherent limitations. Further verification through experiments such as host infection models, gene knockout and complementation is still required.

## Figures and Tables

**Figure 1 pathogens-14-01212-f001:**
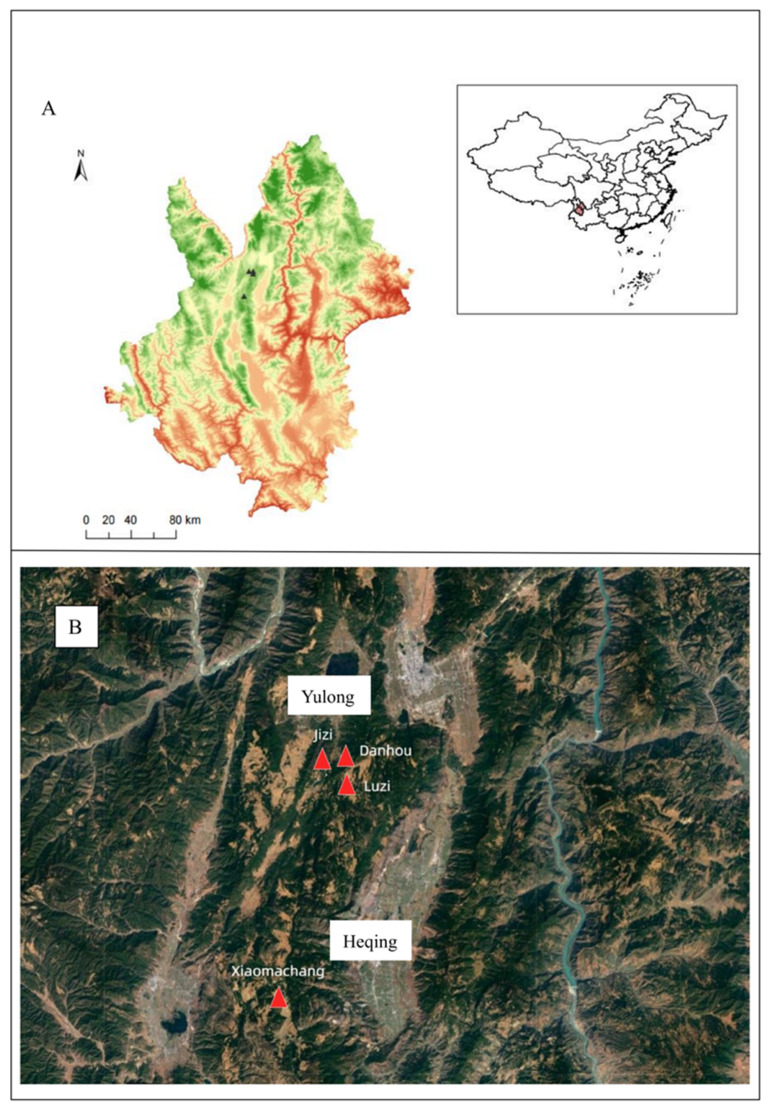
Geographic locations of the isolates in Heqing County and Yulong County. (**A**) The black triangles in the figure indicate the geographic locations of three episodes of sylvatic plague occurred in Yunnan; (**B**) To better delineate the topography of the affected areas and the distances between them, this study employs red triangles to represent the three sylvatic plague outbreak sites.

**Figure 2 pathogens-14-01212-f002:**
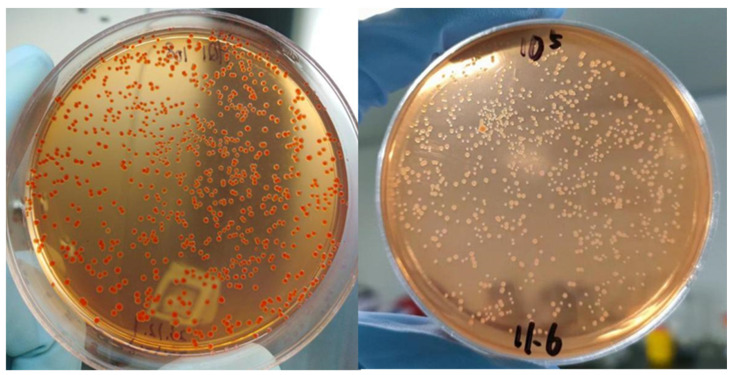
HQ1 (**left**) and EV76 colonies (**right**) on Congo Red culture medium.

**Figure 3 pathogens-14-01212-f003:**
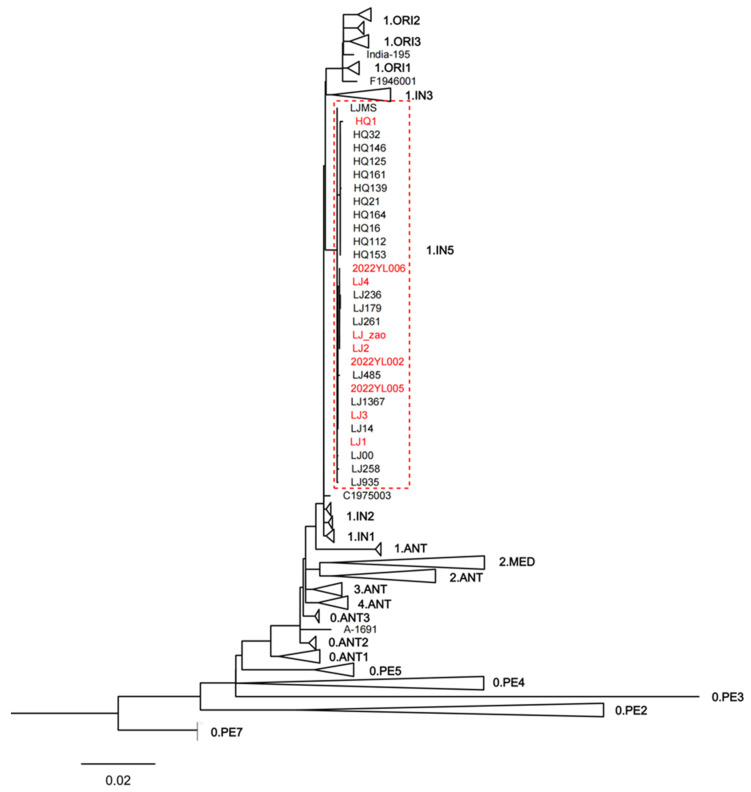
Maximum likelihood phylogenetic tree for the nine newly isolated Yunnan strains along with 234 publicly available sequences. Note: This maximum likelihood phylogenetic tree uses CO92 as the reference strain and isolates from the 0.PE7 lineage as the outgroup. To enhance the clarity of this maximum likelihood tree, information from 243 strains was collapsed and represented by triangles, with each triangle denoting one lineage. To display the internal relationships within the 1.IN5 lineage, this specific lineage was not collapsed. Strain labels highlighted in red correspond to the 9 Yersinia pestis strains isolated from the 2022 Yunnan sylvatic plague.

**Figure 4 pathogens-14-01212-f004:**
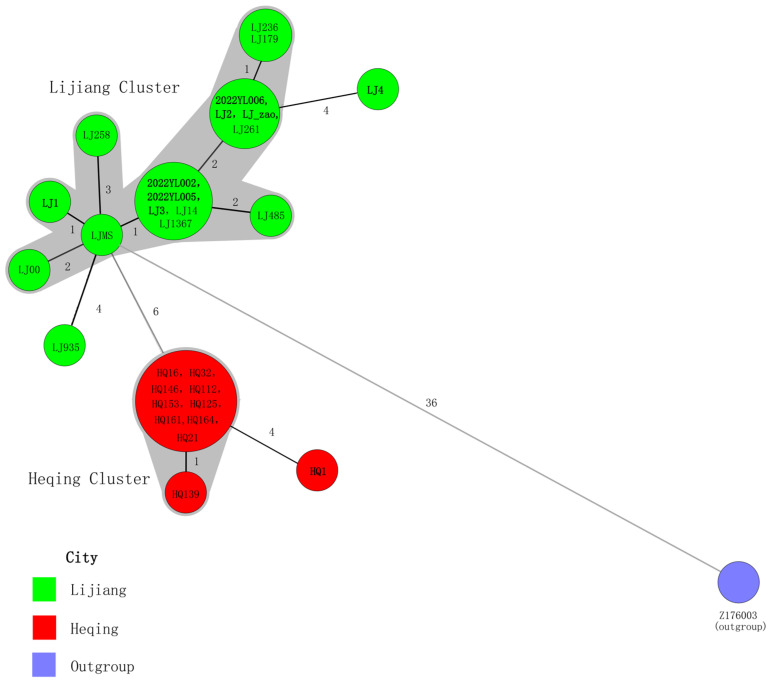
Minimum spanning tree of 29 *Yersinia pestis* strains. Note: A minimum spanning tree was constructed using the SNP genotyping data. The SNP types are displayed as circles, and the size of the circle indicates the number of isolates with the particular type. The numbers marked on the lines between two adjacent circles represent the number of SNP differences. Two adjacent circles that differ at no more than 3 loci are defined as a complex and are highlighted by a shared color halo. The strain numbers in bold font are the Yersinia pestis strains of 2022.

**Figure 5 pathogens-14-01212-f005:**
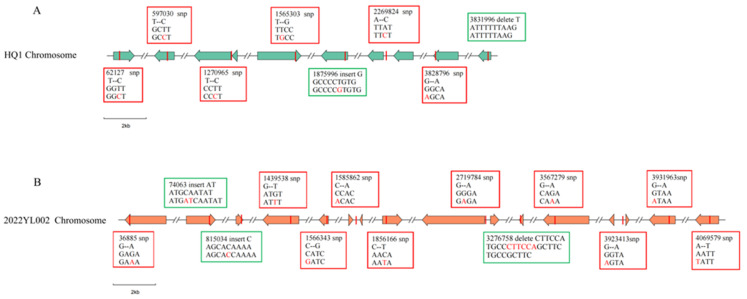
Differences in SNPs and Indels between *Yersinia pestis* strains isolated from wild rodents in Yunnan Province in 2022 and previously isolated strains. A right-pointing arrow corresponds to a gene sequence on the forward DNA strand, and a left-pointing arrow to one on the reverse strand. For clarity, the HQ1 and 2022YL002 chromosomal sequences are assigned different colors. Note: (**A**) Genomic comparison between strains isolated from Heqing in 2022 and previously isolated Heqing strains; (**B**) Genomic comparison between strains isolated from Lijiang in 2022 and previously isolated Lijiang strains. SNPs are highlighted in red boxes, and Indels are marked in green boxes.

**Table 1 pathogens-14-01212-t001:** Background information of the nine *Yersinia pestis* isolates in Yunnan.

Strain ID	Date	City	County	Address	Host	Host State
2022YL002	11 April 2022	Lijiang City	Yulong County	Jizi Village	*R. tanezumi*	dead
2022YL005	12 April 2022	Lijiang City	Yulong County	Jizi Village	*A. chevrieri*	dead
2022YL006	12 April 2022	Lijiang City	Yulong County	Jizi Village	*E. miletus*	dead
HQ1	13 June 2022	Dali Prefecture	Heqing County	Xiaomachang Village	*E. miletus*	live
LJ1	31 October 2022	Lijiang City	Yulong County	Luzi Village	*E. miletus*	dead
LJ2	1 November 2022	Lijiang City	Yulong County	Jizi Village	*E. miletus*	dead
LJ_zao	1 November 2022	Lijiang City	Yulong County	Jizi Village	*Ctenophthalmus quadratus*	——
LJ3	2 November 2022	Lijiang City	Yulong County	Jizi Village	*R. tanezumi*	dead
LJ4	2 November 2022	Lijiang City	Yulong County	Danhou Village	*E. miletus*	dead

**Table 2 pathogens-14-01212-t002:** Genomes and SNPs of the 29 *Yersinia pestis* isolates in Yunnan.

Strain ID	Total Length (bp)	(G + C)%	N50	L50	Contigs	Total Genes	Number of SNP	Synonymous SNPs	Non-synonymous SNPs	CDS	Non-CDS
2022YL002	4656201	47.5%	49295	33	178	4625	71	19	40	59	12
2022YL005	4657164	47.5%	49393	33	185	4648	69	18	39	57	12
2022YL006	4646058	47.5%	48916	33	175	4605	69	17	38	55	14
HQ1	4652251	47.5%	49629	32	175	4633	74	16	41	57	17
LJ1	4655783	47.5%	42693	36	189	4653	70	17	40	57	13
LJ2	4652662	47.5%	48799	33	174	4631	72	17	41	58	14
LJ_zao	4647096	47.5%	49885	32	182	4608	70	14	42	56	14
LJ3	4656336	47.5%	49855	32	179	4608	69	14	41	55	14
LJ4	4644272	47.5%	48799	33	173	4625	74	16	42	58	16
LJ14 *	4663300	47.5%	49244	31	194	4549	67	16	39	55	12
LJ485 *	4664642	47.5%	49368	31	200	4559	68	17	38	55	13
LJ1367 *	4607135	47.5%	48666	32	199	4504	67	15	39	54	13
LJ935 *	4817642	47.6%	4646094	1	4	4604	63	12	38	50	13
LJMS *	4652746	47.5%	48904	32	207	4561	66	14	38	52	14
LJ00 *	4652283	47.5%	48904	32	209	4561	66	16	39	55	11
HQ16 *	4821054	47.6%	4648852	1	4	4609	67	13	39	52	15
HQ21 *	4661100	47.5%	48626	33	218	4567	72	17	41	58	14
HQ32 *	4664294	47.5%	49251	31	205	4568	68	15	40	55	13
HQ112 *	4663758	47.5%	49251	31	203	4566	68	15	40	55	13
HQ146 *	4664356	47.5%	49243	31	205	4560	68	14	41	55	13
HQ153 *	4664890	47.5%	49818	33	208	4566	67	14	41	55	12
HQ161 *	4664293	47.5%	49251	31	205	4556	68	15	39	54	14
HQ164 *	4663823	47.5%	49251	31	204	4546	66	13	40	53	13
HQ125 *	4663671	47.5%	49523	31	200	4553	67	13	40	53	14
HQ139 *	4664804	47.5%	49160	33	210	4564	68	14	41	55	13
LJ179 *	4675707	47.5%	49244	31	211	4571	70	17	39	56	14
LJ236 *	4664450	47.5%	49244	31	197	4566	70	18	39	57	13
LJ258 *	4685025	47.5%	49244	31	222	4574	67	15	38	53	14
LJ261 *	4663094	47.5%	49530	31	192	4539	69	18	39	57	12

Notes: CDS indicates the number of SNPs in the coding region; Non-CDS indicates the number of SNPs not in the coding region. * indicates a strain isolated previously.

**Table 3 pathogens-14-01212-t003:** Comparison of SNP Number Differences.

SNP Category	Number of SNPs (x ± s)	t	*p*
New isolated strain SNPs in 2022	70.89 ± 2.03	4.87	0.001
Previously isolated strain SNPs	67.60 ± 1.85		
New isolated strain Synonymous SNPs in 2022	16.44 ± 1.67	2.51	0.036
Previously isolated strain Synonymous SNPs	15.05 ± 1.73		
New isolated strain Non-synonymous SNPs in 2022	40.44 ± 1.33	2.35	0.047
Previously isolated strain Non-synonymous SNPs	39.40 ± 1.05		
Non-synonymous SNPs/Synonymous SNPs ratio of New isolated strain in 2022	2.46 *	−1.28	0.210
Non-synonymous SNPs/Synonymous SNPs ratio of Previously isolated strain	2.62 *		

Data marked with * was not reported as x ± s.

**Table 4 pathogens-14-01212-t004:** Genomic Variation Analysis of Yersinia pestis Strains from the 2022 Heqing Sylvatic Plague Outbreak in Comparison with Historical Isolates.

Site	Coding Region	Variation Type	Base Change	Encoding	Functional Categories
62127	Yes	Synonymous SNP	T → C	Hypothetical protein	Function unknown
597030	Yes	non-synonymous SNP	T → C	Hydroxylated aromatic compound efflux system	Inorganic ion transport and metabolism
1270965	No	Non-coding region SNP	T → C	—	—
1565303	Yes	Synonymous SNP	T → G	β-N-acetylglucosidase	Carbohydrate transport and metabolism
1875996	Yes	Insertion	Insert G	λ phage Rac integrase	Replication, recombination and repair
2269824	No	Non-coding region SNP	A → C	—	—
3828796	Yes	Synonymous SNP	G → A	Flagellar P-cyclin FlgI	Cell motility
3831996	Yes	Deletion	Delete T	Flagellin FlgF	Cell motility

**Table 5 pathogens-14-01212-t005:** Genomic Variation Analysis of Yersinia pestis Strains from the 2022 Lijiang Sylvatic Plague Outbreak in Comparison with Historical Isolates.

Site	Coding Region	Variation Type	Base Change	Encoding	Functional Categories
36885	Yes	Synonymous SNP	G → A	Methyl-accepting chemotaxis protein	Signal transduction mechanisms
74063	Yes	Insertion	Insert AT	Hypothetical protein	Function unknown
815034	Yes	Insertion	Insert C	ABC transporter substrate-binding protein	Inorganic ion transport and metabolism
1439538	Yes	Synonymous SNP	G → T	2-succinyl-5-enolpyruvyl-6-hydroxy-3-cyclohexene	Coenzyme transport and metabolism
1566343	Yes	non-synonymous SNP	C → G	Iron uptake regulator Fur	Signal transduction mechanisms
1585862	No	Non-coding region SNP	C → A	—	—
1856166	Yes	non-synonymous SNP	C → T	LysR-type transcriptional regulator	Transcription
2719784	No	Non-coding region SNP	G → A	—	—
3276758	Yes	Deletion	Delet CTTCCA	Relevant results are not commented	—
3567279	Yes	non-synonymous SNP	G → A	Hypothetical protein	Function unknown
3923413	No	Non-coding region SNP	G → A	—	—
3931963	Yes	non-synonymous SNP	G → A	Aspartate aminotransferase	Amino acid transport and metabolism
4069579	Yes	non-synonymous SNP	A → T	Pentatricopeptide repeat (PPR) protein family	RNA binding

## Data Availability

The original data presented in the study are openly available in the National Microbiology Data Center at NMDC20418759.

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
