# Peer review of "Molecular Tracing and Comparative Genomics Analysis of *Yersinia pestis* Strains Isolated from Wild Rodents in Yunnan Province in 2022"

_pathogens, 2025, doi:10.3390/pathogens14121212_

Round 1
Reviewer 1 Report
Comments and Suggestions for Authors
The manuscript presents new data on Yersinia pestis strains isolated in 2022 from the endemic 1.IN5 plague focus in Lijiang, Yannan. It demonstrates that these strains are of local origin. The authors conclude that the focus is expanding, which requires its careful monitoring to prevent spillover into human populations. The results of the study are presented clearly, but minor correction of the manuscript is necessary.
- There is no information on the historical strains (20 strains) previously isolated in Lijiang and Heqing with which the 2022 strains are compared. It is necessary to indicate in the text at least the years of isolation of these strains.
- There is no reference in the text to the 7th source in the list of references.
- The phylogenetic relationships of strains within cluster 1.INT5 in Figures 3 and 4 differ from each other. The differentiation of Lijiang strains in cluster 1.INT5 in Figure 3 is poorly visualized, the text lacks a description of this figure. Please provide an explanation of Figure 3 in the text.
- The abstract and section 2.5 indicate that the HQ1 strain differs from Heqing isolates in the past by 6 SNPs, while Figure 4 indicates only 4 SNPs.
- In the introduction and other parts of the manuscript, my text contains some word splicing, which may be due to an incorrect download of the manuscript. Please check the manuscript again. I provide examples of such errors in the Introduction and Materials and Methods sections:
– From April to November 2022, three episodes of sylatic plague occurred sequentially in Yunnan Province (Figure 1A) affecting areas ofmountainous forests and cultivated lands in Jizi Village, Luzi Village, and Danhou Village (Figure 1B).
– Figure 1. Geographic locations of the isolates in Heqing County and Yulong. CountyThis study involved 9 Yersinia pestis strains isolated from the 2022 Yun-nan sylvatic plague outbreak…
– Notably,the primary hosts in the Lijiang sylvatic plague focus…
- From April to November 2022, three episodes of sylatic plague occurred sequentially in Yunnan Province (Figure 1A)
Author Response
4.The abstract and section 2.5 indicate that the HQ1 strain differs from Heqing isolates in the past by 6 SNPs, while Figure 4 indicates only 4 SNPs.
The reference strains used in the two analyses are different. When comparing HQ1 with past Heqing isolates, HQ1 was used as the reference, whereas in Figure 4, Z176003 was used as the reference. This difference in reference strains explains the inconsistency in the reported SNP counts.
The remaining content requiring modification is highlighted in red in the manuscript.

Reviewer 2 Report
Comments and Suggestions for Authors
This manuscript provides valuable genomic insights into the 2022 sylvatic plague outbreaks in Yunnan. The sequencing and phylogenetic analyses are solid and confirm the local persistence of the Y. pestis 1.IN5 lineage. The data are strong, but the interpretation could be expanded, particularly regarding functional and epidemiological implications. With few revisions, this study could make a meaningful contribution to the field of plague surveillance and pathogen genomics.
Comments:
- How do the findings from this study go beyond confirming that the outbreaks came from local plague reservoirs, and what new biological or epidemiological insights can we take away that inform future surveillance or control strategies?
- The study identifies notable mutations, such as the fur non-synonymous SNP and the FlgF deletion, a deeper interpretation of their possible functional impact, perhaps drawing on existing computational predictions or previous experimental work would strengthen the study.
- While the phylogenetic analyses show that the strains belong to the 1.IN5 lineage, do the data allow clearer separation of the three 2022 outbreaks (Heqing vs. Lijiang), and if so, add a comment on this improving our understanding of local transmission dynamics.
- Since the study relies on genomic comparisons without experimental validation, authors should explicitly acknowledge this limitation and outline what kinds of follow-up studies (e.g., virulence assays, host infection models) would be most informative?
- Tables 4–5 would be clearer if key genes were annotated with their functional categories.
- Adding more recent references on plague genomics and surveillance (post-2021) would strengthen the context.
Author Response
- How do the findings from this study go beyond confirming that the outbreaks came from local plague reservoirs, and what new biological or epidemiological insights can we take away that inform future surveillance or control strategies?
The 2022 Yunnan wild rodent plague exhibited the following characteristics: first, it formed a stable clone population in a relatively isolated area; second, it demonstrated two epidemic patterns (infected animals either dying from the disease or acting as asymptomatic carriers); and third, it could cross species barriers to infect domestic rodents. Based on these findings, the proposed scientific prevention and control strategy involves continuous monitoring, heightened vigilance toward asymptomatic carrier rodents, and sustained efforts to manage domestic rodent populations.
- The study identifies notable mutations, such as the fur non-synonymous SNP and the FlgF deletion, a deeper interpretation of their possible functional impact, perhaps drawing on existing computational predictions or previous experimental work would strengthen the study.
Non-synonymous SNPs in the fur gene were predicted to affect its function, but the specific effects require experimental validation by comparing iron uptake regulation between strains with and without the mutation. However, we currently lack a mature experimental protocol. Regarding the FlgF deletion, since Yersinia pestis has lost motility and does not express the flagellar protein FlgF, investigating its impact on the virulence of Y. pestis is not considered worthwhile.We are grateful for these constructive suggestions and will incorporate them into subsequent studies.
- While the phylogenetic analyses show that the strains belong to the 1.IN5 lineage, do the data allow clearer separation of the three 2022 outbreaks (Heqing vs. Lijiang), and if so, add a comment on this improving our understanding of local transmission dynamics.
We thank the reviewer for this insightful question. Our data do indeed allow for clearer discrimination among these three outbreaks. The minimum spanning tree constructed based on whole-genome SNP analysis revealed that the strains from the Heqing outbreak formed a distinct cluster (Heqing Cluster), while the two Lijiang outbreaks, despite their temporal separation, collectively belonged to a separate independent branch (Lijiang Cluster). Finer resolution of SNP genotypes showed that the three epidemic sites (Jizi, Luzi and Danhou) exhibited distinct genotypes. Notably, strains from both Lijiang outbreaks were detected at the "Jizi" site and includeded two genotypes, with each genotype containing strains from both outbreaks.
The remaining content requiring modification is highlighted in blue in the manuscript.

Reviewer 3 Report
Comments and Suggestions for Authors
The manuscript “Molecular Tracing and Comparative Genomics Analysis of Yersinia pestis Strains Isolated from Wild Rodents in Yunnan Province in 2022” by Rong Yang and coauthors conducted a genomic study of nine Y. pestis isolates from recent outbreaks of sylvatic plague in rodents in China. The study provides important genomic data on the latest epizootic outbreaks in China and discusses their risk and importance to human health. However, there are several methodological and argumentative aspects that need to be clarified and/or corrected to justify the publication of the study.
• Manuscript Type: Given the current approach, I don't think it should be categorized as a "viewpoint." Instead, I suggest reclassifying it as an "Article" or "Communication."
• Manuscript Structure: Renumber the manuscript sections, starting with the Introduction (1), and I suggest including a conclusions section.
• Abstract: It is very extensive and should be limited to a maximum of 200 words according to the journal's guidelines.
• Introduction: I recommend including a brief background of the Y. pestis lineages 0.PE, ORI, MED, ANT and locating where the latest outbreaks of the 21st century in China are located (lineage, sublineage).
• Section 1.1: Correct the text at the beginning (move the word "county" to the legend of Figure 1).
• Section 1.2: The authors mention the use of a BSL-3 laboratory as early as 1.3. However, it should be mentioned even earlier, in this section. Also, should reference [7] be included in this section?
• Section 1.4.1: How was the assembly performed (de novo?, by reference?, with which assembler?)
• Section 1.4.2: The authors assign this section to 3rd generation sequencing (nanopore? PacBio?), however the procedures are essentially specific to 2nd generation (i.e. Illumina). This is a significant methodological error. Should the authors clarify which sequencing procedures were actually performed?
• Sections 1.5-1.9: It's not clear what assembly they used. 2nd generation, 3rd generation, or hybrid assembly?
• Section 1.8: Indicate the version and reference of VarScan software.
• Figure 5: I suggest merging the SNPS information from this figure as a column within Tables 4 and 5.
• Discussion: In 3.2, I suggest contrasting the stability and evolutionary rate of Y. pestis SNPs based on references (i.e. https://doi.org/10.1038/ng.705, https://doi.org/10.1073/pnas.1205750110).
• Language: There are some errors and mistyping throughout the manuscript. I suggest reviewing it with a native speaker.
• Language: There are some errors and mistyping throughout the manuscript. I suggest reviewing it with a native speaker.
Author Response
Discussion: In 3.2, I suggest contrasting the stability and evolutionary rate of Y. pestis SNPs based on references (i.e. https://doi.org/10.1038/ng.705, https://doi.org/10.1073/pnas.1205750110).
Thank you for this valuable suggestion. Comparing our findings with those of peer experts through a comparative study is indeed an insightful perspective. However, in our current research, we have primarily focused on SNP identification, the construction of phylogenetic and minimum spanning trees, a quantitative comparison of synonymous and non-synonymous SNPs, and comparative genomic analyses. We have not yet delved into investigating the stability and evolutionary rates of the SNPs. Furthermore, current methodological constraints in our experimental setup presently prevent us from addressing this specific aspect of the research. We sincerely appreciate your suggestion once again.
The remaining content requiring modification is highlighted in orange in the manuscript.

Round 2
Reviewer 2 Report
Comments and Suggestions for Authors
The authors have done an excellent job revising the manuscript. They have addressed all my comments thoroughly and efficiently. The manuscript is now ready for publication, and I congratulate the authors on a well-executed and impactful study.
Reviewer 3 Report
Comments and Suggestions for Authors
No further coments.